# MME-CC: A Challenging Multi-Modal Evaluation Benchmark of Cognitive Capacity

## Abstract

As reasoning models scale rapidly, the essential role of multimodality in human cognition has come into sharp relief, driving a growing need to probe vision-centric cognitive behaviors. Yet, existing multimodal benchmarks either overemphasize textual reasoning or fall short of systematically capturing vision-centric cognitive behaviors, leaving the cognitive capacity of MLLMs insufficiently assessed. To address this limitation, we introduce MME-CC (**M**ulti-**M**odal **E**valuation benchmark of **C**ognitive **C**apacity), a vision-grounded benchmark that organizes 11 representative reasoning tasks into three fundamental categories of visual information—spatial, geometric, and knowledge-based reasoning—and provides fine-grained analyses of MLLMs' cognitive capacity across these dimensions. Based on MME-CC, we conduct extensive experiments over 16 representative MLLMs. Our study reveals that closed-source models currently lead overall (e.g., 42.66 for Gemini-2.5-Pro vs. 30.45 for GLM-4.5V), while spatial and geometric reasoning remain broadly weak ($\leq 30\%$). We further identify common error patterns—including orientation mistakes, fragile cross-view identity persistence, and poor adherence to counterfactual instructions—and observe that Chain-of-Thought typically follows a three-stage process (extract → reason → verify) with heavy reliance on visual extraction. We hope this work catalyzes a shift toward treating the cognitive capacity of MLLMs as central to both evaluation and model design.

## 1 Introduction

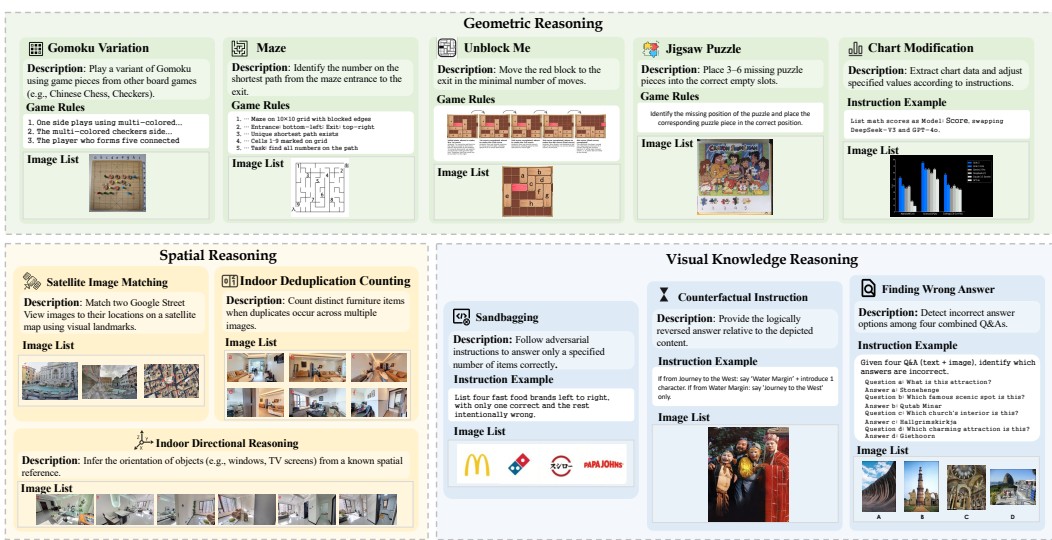

Figure 1: **Task taxonomy of MME-CC.** Three major task categories are defined—Spatial Reasoning, Geometric Reasoning, and Visual Knowledge Reasoning—each with representative subtasks and one illustrative input example.

Vision is a crucial means for humans to perceive the world. Naturally, multimodal large language models (MLLMs) have become a key research direction for researchers in their pursuit of Artificial General Intelligence (AGI). Various analytical studies focused on multimodal understanding are thriving, aiming to fully uncover the potential flaws of MLLMs and guide the iteration of them. A growing number of benchmarks have been introduced to evaluate multimodal language models (MLLMs) across diverse visual reasoning tasks, encompassing general image understanding (Liu et al., 2024b; Fu et al., 2023), multi-disciplinary multimodal reasoning (Yue et al., 2024; Lu et al., 2024; Yue et al., 2025), and open-domain scenarios (xAI, 2024; He et al., 2024; Liu et al., 2024a). Notably, state-of-the-art MLLMs often demonstrate strong performance on these benchmarks.

While a bunch of MLLM benchmarks claim to evaluate the cognitive capacity of MLLMs, they have various flaws. These flaws make them either lean too much toward textual capabilities or lack sufficient coverage of vision-based cognitive behaviors when assessing MLLMs' cognitive capacity. MathVista (Lu et al., 2024) and MMMU Series (Yue et al., 2024; 2025) are overly biased toward the text-space-based reasoning capabilities of MLLMs. On the other hand, other benchmarks (e.g. ZeroBench (Roberts et al., 2025)) measure the cognitive capacity of MLLM by enumerating various vision-based reasoning tasks, but lack a well-established classification system and in-depth analysis of the cognitive capacity of MLLM.

To remedy the gap, we introduce MME-CC (Multi-Modal Evaluation Benchmark of Cognitive Capacity), a benchmark of vision-based cognitive tasks for MLLMs. Specifically, MME-CC firstly introduces a set of vision-based cognitive tasks that examine three essential dimensions of MLLMs' reasoning: Spatial Information, Geometric Information, and Visual Knowledge Reasoning. In addition, these tasks are organized into 11 distinct representative visual reasoning problems, as shown in Figure 1, each attributed to one of the three dimensions. Moreover, built with the extensive efforts of a 10-person annotation team, including dedicated subtask leads and a task lead, MME-CC underwent multiple stages of human review and cross-checking to ensure validity. On this basis, using 1,173 questions carefully annotated by human experts, MME-CC conducts a detailed analysis of the current cognitive capacity of MLLMs. Finally, MME-CC also reveals the strengths and weaknesses, Chain-of-Thought (CoT) patterns, and error patterns of 16 representative MLLMs in different dimensions of vision-based cognitive capacity.

Our contributions are as follows:

- We provide **MME-CC**, a high-quality, language-independent visual reasoning benchmark that fills the gap in MLLM cognitive capacity categorization and systematic analysis.
- **We uncover several key insights into current MLLM cognitive capacity:**
  - Closed-source models consistently outperform open-source counterparts, with Gemini-2.5-Pro achieving the best performance (42.66) compared to the strongest open-source model, GLM-4.5V (30.45).
  - Spatial and geometric reasoning remain broadly weak, with both categories scoring at or below 30%.
  - Common error patterns include orientation and reference-frame mistakes, poor cross-view identity persistence, and limited adherence to counterfactual instructions.
  - CoT reasoning typically follows a three-stage layered process—extraction, reasoning, and verification—with visual extraction involved throughout.

## 2 MME-CC

We construct MME-CC benchmark to systematically evaluate the visual cognitive and reasoning abilities of MLLMs. This benchmark is designed to facilitate a comprehensive and rigorous assessment of model capabilities through a series of meticulously crafted tasks. The core of this benchmark comprises three primary task categories: Spatial Reasoning, Geometric Reasoning, and Visual Knowledge Reasoning, which respectively serve to examine model performance within each corresponding dimension.

### 2.1 DATA CONSTRUCTION PIPELINE

As illustrated in Figure 2, our human-in-the-loop pipeline produces high-quality evaluation data through iterative definition, acquisition, processing, and filtering.

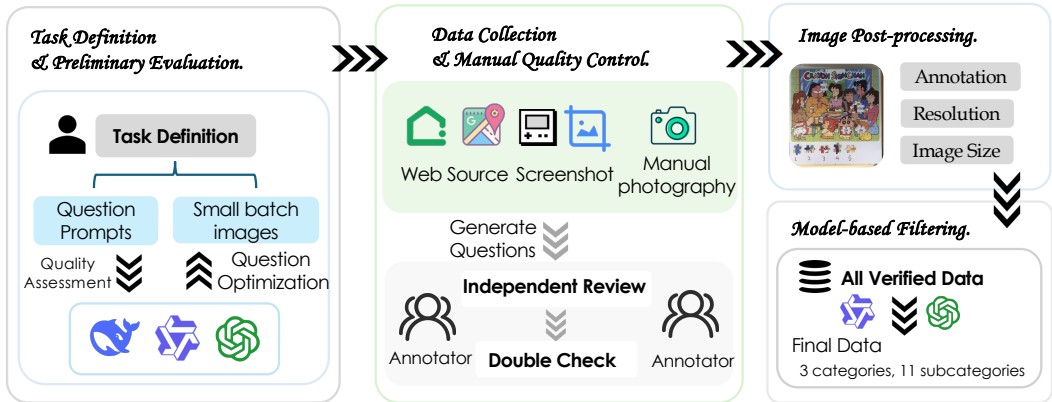

Figure 2: **Overview of the data construction and quality control pipeline.** The pipeline comprises four stages: (1) *Task definition and preliminary evaluation* — subtasks are defined with clear objectives, and small-scale pilots are conducted to validate prompt design and calibrate difficulty; (2) *Data acquisition and manual verification* — images from license-compliant sources are annotated and cross-checked to ensure quality; (3) *Post-processing* — standardized procedures (e.g., cropping, resolution checks, identifier assignment) are applied to unify formatting; (4) *Model-based filtering* — items that are overly simple, redundant, or ambiguous are removed based on MLLM performance, and the remaining samples form the final benchmark.

**Annotators.** We employed a structured 10-person team (6 annotators, 3 subtask leads, 1 task lead), all with prior multimodal evaluation experience. All annotators were required to review a detailed instruction manual and pass a corresponding qualification test before commencing the annotation tasks (see Appendix D for the full guidelines). Each subtask had a dedicated lead responsible for design, pipeline coordination, and multi-stage quality review, ensuring consistent annotation and rigorous quality control across the data lifecycle.

Table 1: Annotator roles and responsibilities for the evaluation set (N = 10).

| Participant | Role | Job Description |
|---|---|---|
| *A* Annotators | A1–A6 | Question construction and multi-round quality checks |
| *B* Subtask Leads | B1–B3 | Subtask design and end-to-end oversight; with over 3 years of experience in relevant fields (e.g., NLP, CV) and prior work on dataset creation. |
| *C* Task Lead | C1 | Overall benchmark owner; extensive evaluation experience; with a documented history of leading the development of 3+ public benchmarks. |

**Task Definition & Preliminary Evaluation** Each subtask has a clear objective and a concise, task-specific prompt template. To validate the design, we built a pilot set of 5–10 examples from carefully selected images for each subtask. Rather than ad-hoc sampling, construction followed a structured design process specifying evaluation dimensions and steps. We piloted with two models (Doubao and Gemini) to test clarity and feasibility, then iteratively refined the task scope, prompts, and data handling. Reliability was further ensured through explicit controls. Detailed construction procedures and reliability controls for each task are documented in Appendix C.

**Data Collection.** After definitions stabilize, we collect and annotate images from diverse, license-compliant sources:

(1) Real-world photographs, such as board games (e.g., Gomoku) and puzzle scenes.
(2) Targeted screen captures from online platforms, including comics, real-estate listings (e.g., Lianjia), Google Street View, and video content (e.g., YouTube, Bilibili).
(3) Game screenshots that cover representative in-game reasoning cases.
(4) Additional types of images as required by specific subtasks.

Table 2: **Task taxonomy and dataset statistics of MME-CC.** "Source" indicates the data origin, "#Q" is the number of samples, and "I."/"O." represent the average input and output token lengths measured with `Doubao-Seed-1.6-vision-0815`.

| Task | Description | Source | #Q | I. | O. |
|---|---|---|---|---|---|
| **Spatial Reasoning** | | | *(3 tasks, 319 samples)* | | |
| Satellite Image Matching | Match street view images to their satellite map locations using visual clues. | Google Maps | 101 | 4,335 | 2,768 |
| Indoor Directional Reasoning | Determine object orientations using spatial reference points. | Real Estate | 113 | 10,168 | 5,090 |
| Indoor Deduplication Counting | Count unique furniture instances across multiple views. | Real Estate | 105 | 10,091 | 4,370 |
| | *Total / Avg.* | | **319** | **8,198** | **4,076** |
| **Geometric Reasoning** | | | *(5 tasks, 605 samples)* | | |
| Gomoku Variation | Solve visual logic problems based on Gomoku with mixed game pieces. | Photography | 122 | 1,411 | 4,700 |
| Unblock Me | Find the shortest move sequence to unblock a target block. | Game Screenshots | 99 | 3,483 | 11,741 |
| Maze | Identify the correct number on the shortest path in a maze. | Auto-generated | 194 | 336 | 9,755 |
| Jigsaw Puzzle | Complete missing puzzle pieces in a partial layout. | Photography | 141 | 1,765 | 2,330 |
| Chart Modification | Modify chart values based on instructions. | Web Images | 49 | 751 | 2,497 |
| | *Total / Avg.* | | **605** | **1,549** | **6,204** |
| **Visual Knowledge Reasoning** | | | *(3 tasks, 249 samples)* | | |
| Sandbagging | Follow adversarial instructions to answer partially correctly. | Web Images | 41 | 1,753 | 672 |
| Counterfactual Instruction | Provide the reversed answer based on depicted facts. | Web Images | 60 | 1,379 | 625 |
| Finding Wrong Answer | Detect incorrect answer choices from grouped Q&As. | Internal Dataset | 148 | 5,122 | 2,691 |
| | *Total / Avg.* | | **249** | **2,751** | **1,329** |
| **Total** | | | **1,173** | **4,166** | **3,904** |

**Image Post-processing & Model-based Filtering.** All collected images undergo a unified post-processing pipeline, including ID assignment, cropping, and related adjustments (see Appendix C). We then apply leading models (e.g., Gemini-2.5-Pro) to filter the data. This process removes items that are trivially easy (defined as achieving a model accuracy score above 95%, semantically redundant, or lacking discriminative value. In total, this filtering stage removed approximately 50% of the initial data pool. The remaining pool serves as the foundation for constructing the final benchmark set used in downstream evaluation.

## 2.2 Data Quality Assurance

In addition to the quality controls embedded throughout the data collection process, the finalized dataset undergoes the following further checks:

**Manual Verification.** Each sample is double-checked: one annotator provides the reference answer and another independently verifies it. In cases of disagreement, the sample was escalated to the respective subtask lead for a final binding decision. Every item undergoes at least two rounds of review, and a dedicated QA team conducts periodic audits to ensure accuracy and consistency.

**Exception Handling.** Samples that yield zero accuracy across screened models are randomly reviewed to identify potential annotation issues or hidden shortcuts; problematic items are corrected or removed.

Table 3: **Comparison of representative benchmarks.** "Vision-based" indicates that all task information is derived from images rather than text, "Output" specifies the answer format (MC = multiple-choice, FF = free-form), and "Source" denotes the dataset origin.

| Benchmark | Input | Vision-based | Output | Source |
|---|---|---|---|---|
| MME (Fu et al., 2023) | 1 Image | ✗ | MC | Existing |
| MMMU (Yue et al., 2024) | ≥1 Image | ✗ | MC / FF | Diverse |
| MMBench (Liu et al., 2024b) | 1 Image | ✗ | MC | Diverse |
| MMStar (Chen et al., 2024) | 1 Image | ✗ | MC | Existing |
| Zerobench (Roberts et al., 2025) | ≥1 Image | ✓ | FF | Diverse |
| **MME-CC (Ours)** | ≥1 Image | ✓ | FF | Diverse |

## 2.3 BENCHMARK STATISTICS

Table 2 summarizes the taxonomy and statistics of MME-CC, which organizes 11 subtasks into three reasoning categories: *Spatial*, *Geometric*, and *Visual Knowledge Reasoning*. For each subtask, it reports the data source, sample size, and the average input and output token lengths, where the input length includes both text and image tokens extracted by `Doubao-Seed-1.6-vision-0815`. These statistics indicate the reasoning complexity, as longer sequences require deeper inference, and the considerable output length reflects the need for non-trivial reasoning chains.

In addition, Table 3 compares MME-CC with representative benchmarks. Notably, MME-CC emphasizes vision-based reasoning, since the textual input does not contain any task-specific solution information, thereby ensuring that solving relies primarily on visual understanding and reasoning.

## 3 EXPERIMENTS

### 3.1 EXPERIMENTAL SETUP

**Model Configuration.** We evaluate a range of large language and vision–language models, including both proprietary and open-source systems. For proprietary models, we use the official inference APIs with default settings. For open-source models, we adopt a unified decoding configuration with temperature set to 1.0 and top-p to 0.7, while all other hyperparameters follow their respective defaults.

**Evaluation Metrics.** We employ an LLM-as-a-judge protocol, wherein a language model is prompted to compare the model-generated response against a gold reference and assign a correctness score. Specifically, we adopt `DeepSeek-V3-0324` as the judge model. To verify its reliability, we conduct a manual evaluation of 99 randomly sampled items (33 from each category), achieving a scoring agreement rate of 95% with human judgments. The scoring prompt used for this evaluation is provided in the Appendix E. For each question, `DeepSeek-V3-0324` compares the final answer with the reference and outputs a score in $\{0, 1\}$.

### 3.2 MAIN RESULTS

We conduct a comprehensive evaluation of 16 MLLMs on MME-CC, covering three core reasoning dimensions: *Spatial Reasoning*, *Geometric Reasoning*, and *Visual Knowledge Reasoning*. The results are shown in Table 4.

**MLLMs remain limited in visually grounded reasoning tasks.** The state-of-the-art model Gemini-2.5-Pro achieves an overall accuracy of only 42.66% on MME-CC, with a score of 74.63% in VKR tasks, while its performance in Geometric Reasoning and Spatial Reasoning remains below 30%. These results suggest that current models lack the ability to conduct comprehensive and fine-grained reasoning under purely visual inputs. Their better performance in basic perceptual tasks, such as entity recognition or object detection (e.g., VKR), mainly results from the fact that these tasks are well covered during training, whereas the complex tasks we design require more advanced spatial and geometric reasoning and therefore expose clear limitations.

**Reasoning-oriented models exhibit advantages over non-reasoning models.** In Spatial and Geometric Reasoning tasks, reasoning-oriented models consistently outperform their non-reasoning counterparts. Notably, within the GPT series, GPT-5 (high) achieves superior performance com-

Table 4: **Results on the MME-CC** benchmark across *three* core dimensions: Spatial Reasoning (SR), Geometric Reasoning (GR), and Visual Knowledge Reasoning (VKR). Shaded entries indicate the best performance, **bold** the second-best, and underlined the third-best in each column.

| Model | Reasoning | Overall | SR | GR | VKR |
|---|---|---|---|---|---|
| **Human**[1] (n=99, sampled) | – | 95.86 | 95.83 | 95.83 | 95.92 |
| **Closed-Source Models** | | | | | |
| Gemini-2.5-Pro (Google, 2025b) | ✓ | 42.66 | 23.80 | **29.56** | 74.63 |
| GPT-5 (high) (OpenAI, 2024a) | ✓ | **40.25** | 30.63 | 23.64 | 66.47 |
| Doubao-Seed-1.6-vision-0815 (Think) (ByteDance, 2025) | ✓ | 40.08 | 22.03 | 31.50 | 66.70 |
| Gemini-2.5-Flash (Google, 2025a) | ✓ | 37.57 | 18.60 | 21.21 | **72.90** |
| o4-mini (high) (OpenAI, 2025b) | ✓ | 35.00 | 25.00 | 21.96 | 58.03 |
| GPT-4.1 (OpenAI, 2025a) | ✗ | 32.14 | **27.90** | 12.22 | 56.30 |
| GPT-4o-1120 (OpenAI, 2024b) | ✗ | 26.88 | 22.60 | 10.12 | 47.93 |
| Doubao-Seed-1.6-vision-0815 (Nonthink) (ByteDance, 2025) | ✗ | 25.96 | 23.63 | 23.82 | 30.43 |
| **Open-Source Models** | | | | | |
| GLM-4.5V (Zai-org, 2025b) | ✓ | 30.45 | 13.27 | 13.34 | 64.73 |
| Qwen2.5-VL-72B-Instruct (Qwen Team, 2025b) | ✗ | 23.59 | 12.47 | 8.96 | 49.33 |
| MiMo-VL-7B-RL (MiMo Team, 2025) | ✓ | 20.90 | 9.30 | 11.10 | 42.30 |
| GLM-4.1V-9B-Thinking (Zai-org, 2025a) | ✓ | 19.30 | 8.73 | 9.22 | 39.93 |
| Qwen2.5-VL-32B-Instruct (Qwen Team, 2025a) | ✗ | 14.39 | 9.03 | 8.56 | 25.57 |
| InternVL3-8B (OpenGVLab, 2025) | ✗ | 11.36 | 7.30 | 2.38 | 24.40 |
| Keye-VL-8B-Preview (Kwai-Keye, 2025) | ✗ | 9.65 | 7.70 | 2.04 | 19.20 |
| Qwen2.5-VL-7B-Instruct (Qwen Team, 2025c) | ✗ | 7.50 | 4.70 | 3.22 | 14.57 |

pared to GPT-4.1 across SR and GR tasks. This trend indicates that longer Chain-of-Thought reasoning chains provide additional opportunities for iterative verification of recognition outcomes and intermediate inferences, thereby contributing to improved problem-solving in complex scenarios. A more detailed analysis of this phenomenon is presented in the Discussion section.

**Scaling laws remain valid for visual reasoning tasks.** Within the Qwen2.5 family, performance improves consistently as the parameter scale increases from 7B to 32B and 72B. This observation indicates that complex visual perception and reasoning tasks require broader knowledge capacity to support effective inference, while smaller-scale models face inherent limitations in achievable performance.

## 4 DISCUSSION

Based on the results presented in Table 4, this paper discusses the following research questions:

RQ1: **What are the differences in model performance?**

RQ2: **How does the model reason in visual tasks?**

RQ3: **What error patterns do models exhibit in visual reasoning?**

---

[1]5 students who did not participate in the question setting, with higher degrees

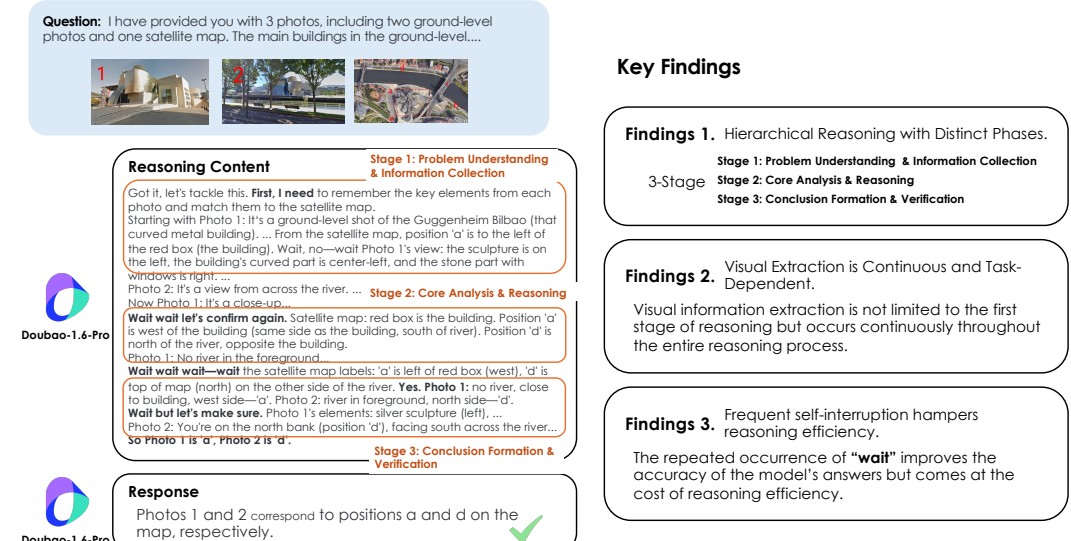

Figure 3: **Detailed CoT analysis of `Doubao-Seed-1.6-vision-0815` on the *Satellite Image Matching* task.** The analysis reveals three key findings: (1) hierarchical reasoning with distinct phases, (2) continuous and task-dependent visual extraction, and (3) frequent self-interruptions that reduce reasoning efficiency.

## 4.1 RQ1: MODELS EXCEL AT DIFFERENT TASKS, YET OVERALL PERFORMANCE REMAINS UNSATISFACTORY

Notably, GPT-5 (high) achieves the highest performance in Spatial Reasoning with a score of 30.3%, a result that appears attributable to its capabilities in sub-tasks requiring complex spatial orientation and object counting (e.g., Indoor Directional Reasoning and Indoor Deduplication Counting). Gemini-2.5-Pro, in contrast, demonstrates a clear advantage in Visual Knowledge Reasoning by attaining a leading score of 70.7%. In the domain of Geometric Reasoning, the Doubao model's performance advantage appears localized to the Jigsaw Puzzle task, whereas Gemini-2.5-Pro shows more robust results across other geometric reasoning challenges (Table 4).

## 4.2 RQ2: MLLMs REMAIN FAR FROM "THINKING LIKE HUMANS"

We analyze the chain-of-thought (CoT) behavior of `Doubao-Seed-1.6-vision-0815` on MME-CC and summarize the following observations.

**Layered, stage-wise reasoning**  The reasoning chain follows three stages: **Stage 1: Problem Understanding & Information Collection**, where the model reads the prompt, scans the image for key objects and relations, and restates the goal; **Stage 2: Core Analysis & Reasoning**, where it proposes options, checks them against visual evidence and rules, and updates assumptions; and **Stage 3: Conclusion Formation & Verification**, where it assembles the confirmed evidence, gives an answer with a brief rationale, and performs a final consistency check. Although task-specific tactics vary, the overall structure remains stable, as shown in Figure 3.

**Image revisiting throughout the process**  Visual information extraction is not confined to the beginning but occurs as needed throughout the reasoning process. In *Satellite Image Matching*, for example, the model repeatedly inspects the original images to re-check building orientation and relative layout, thereby revising earlier spatial judgments.

**Excessive verification reduces efficiency**  The model frequently employs "wait" style pauses for reflection and re-checks. Moderate pausing can reduce errors; excessive pausing, however, leads to stalling and repetitive verification, which is particularly evident in complex spatial relations. On MME-CC spatial and geometric tasks, most models achieve only 20%-30%; in *Maze*, which requires

Table 5: **Performance on MME-CC subtasks.** Each cell reports the ablation score; in parentheses we list the main-experiment score and the signed difference ($\pm\Delta$) from the base, i.e., *ablation = main* $\pm\Delta$. For ablations, we include only subtasks without instruction conflicts with the experimental variable; conflicted subtasks (e.g., Chart Modification, Visual Knowledge Reasoning) are omitted.

| Task | Gemini-2.5-Pro | Doubao-Seed-1.6-vision-0815 | o4-mini-high |
|---|---|---|---|
| **Spatial Reasoning** | | | |
| Satellite Image Matching | 30.5 (28.3 + 2.2) | 41.0 (39.6 + 1.4) | 31.9 (30.3 + 1.6) |
| Indoor Directional Reasoning | 15.4 (14.3 + 1.1) | 4.8 (5.7 −0.9) | 11.2 (11.2 + 0.0) |
| Indoor Deduplication Counting | 29.1 (28.8 + 0.3) | 19.0 (20.8 −1.8) | 33.2 (33.5 −0.3) |
| *Average (Spatial)* | 25.0 (23.8 + 1.2) | 21.6 (22.0 −0.4) | 25.4 (25.0 + 0.4) |
| **Geometric Reasoning** | | | |
| Maze | 1.9 (1.1 + 0.9) | 0.6 (0.8 −0.2) | 1.5 (1.2 + 0.3) |
| Gomoku Variation | 31.0 (34.8 −3.8) | 16.7 (14.9 + 1.8) | 12.5 (12.0 + 0.5) |
| Jigsaw Puzzle | 30.8 (30.4 + 0.4) | 72.1 (70.6 + 1.5) | 26.7 (27.0 −0.3) |
| Unblock Me | 27.7 (26.8 + 0.9) | 29.6 (28.8 + 0.8) | 21.6 (21.4 + 0.2) |
| *Average (Geometric)* | 22.9 (23.3 −0.4) | 29.8 (28.8 + 1.0) | 15.6 (15.4 + 0.2) |
| **Overall Average** | **23.9 (23.5 +0.4)** | **25.7 (25.4 +0.3)** | **20.5 (20.2 +0.3)** |

continued rule-based simulation and path planning, no model exceeds 2%. We hypothesize that long reasoning chains dilute attention, obscure crucial visual details, and ultimately degrade outcomes.

**Textual guidance yields consistent gains** To address this, we add the instruction "You should first describe the relevant content in the image according to the prompt, and then answer the question." As shown in Table 5, most tasks exhibit consistent improvements. The gains indicate that an initial textual description stabilizes subsequent reasoning by anchoring visual perception; in addition, the improvements mainly arise from better textual alignment rather than stronger intrinsic visual reasoning.

### 4.3 RQ3: MLLMs EXHIBIT RECURRING FAILURES IN ORIENTATION JUDGMENT, ENTITY CONSISTENCY, AND INSTRUCTION FOLLOWING

We analyze failure cases in MME-CC and observe several recurring errors that appear across tasks and reasoning dimensions.

**Orientation judgment and reference-frame alignment.** The model often fails to preserve object orientation across views, and viewpoint changes induce mismatches that hinder the establishment of a consistent global reference frame, thus producing orientation errors during reasoning; the issue is notably salient in tasks that require reasoning over indoor layouts or aligning orientations across multiple views, as shown in Figure 4a.

**Entity identity consistency under multi-view settings.** When reasoning over multiple views, the model frequently fails to maintain identity consistency for the same entity in the scene, which leads to double counting or omission, as illustrated in Figure 4b.

**Over-reliance on literal descriptions under instruction constraints.** Faced with non-literal or counterfactual instructions, the model tends to prioritize the literal visual content while ignoring task-specific counterfactual constraints expressed in text, thereby producing answers that conflict with the required instruction, as shown in Figure 14.

Further analyses and complete error cases are provided in Appendix G.

## 5 RELATED WORK

**General multimodal benchmarks.** A broad line of benchmarks evaluates VLMs on perception and language-mediated reasoning. MMBench (Liu et al., 2024b), SEED-Bench (Li et al., 2024), and MMStar (Chen et al., 2024) provide large-scale multiple-choice evaluation with fine-grained skill categorization. MMMU (Yue et al., 2024), MathVista (Lu et al., 2024), and MMMU-Pro (Yue et al.,

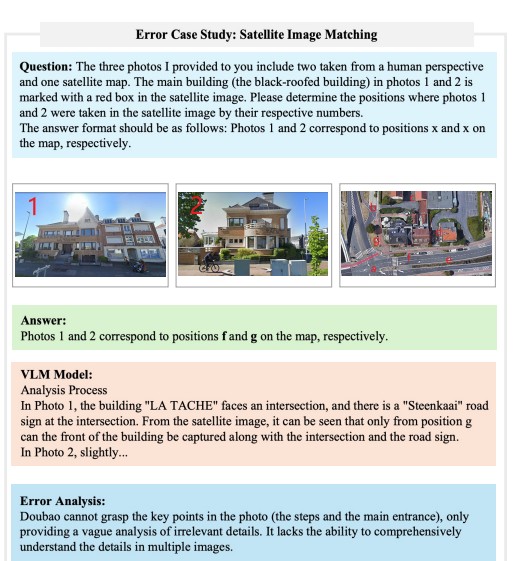

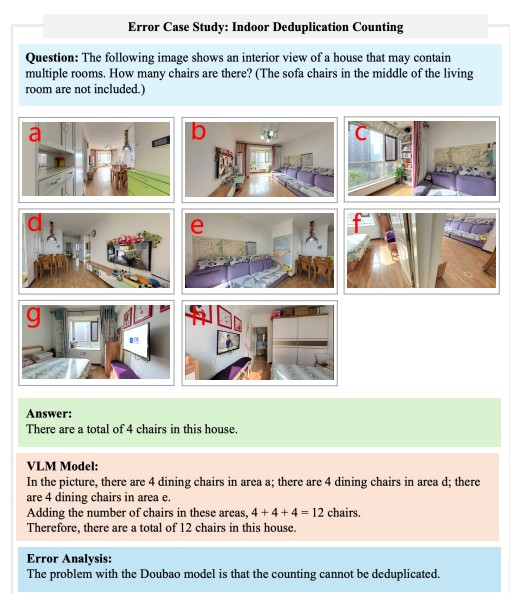

(a) In the *Satellite Image Matching* task, the model fails to capture critical visual cues (e.g., steps and entrance), focusing instead on irrelevant details.

(b) In the *Indoor Deduplication Counting* task, the model fails to deduplicate entities, leading to redundant counting.

Figure 4: Representative error cases of `Doubao-Seed-1.6-vision-0815`.

2025) target multi-disciplinary reasoning across STEM and humanities. Open-domain settings such as RealWorldQA (xAI, 2024), OlympiadBench (He et al., 2024), and VisualWebBench (Liu et al., 2024a) broaden task diversity. While these resources are valuable for overall capability profiling, many items permit solutions that rely on textual cues, format priors, or OCR, which makes it difficult to isolate visual reasoning.

**Language-independent visual reasoning.** Recent analyses report shortcut use in multimodal evaluations, where models exploit answer-bearing text instead of reasoning over images (e.g., NaturalBench, EasyARC, VLSBench). To better probe visual inference, several works explore spatial relations and visual puzzles. ZeroBench (Roberts et al., 2025) stresses spatial and commonsense limits with carefully designed queries, and VisuLogic (Xu et al., 2025) offers human-verified problems spanning spatial relations, geometric abstraction, and visual planning. However, many existing tasks are constrained by synthetic data, repetitive templates, or narrow formats, which limits novelty and reduces the headroom for assessing language-independent reasoning.

## 6 CONCLUSION

We present MME-CC as a vision-grounded benchmark that organizes eleven representative tasks into spatial, geometric, and visual-knowledge dimensions, and we provide fine-grained analyses of multimodal models' cognitive capacity across these dimensions. We evaluate sixteen representative models and observe that closed-source systems currently lead overall (42.66 for Gemini-2.5-Pro vs. 30.45 for GLM-4.5V), while spatial and geometric reasoning remain comparatively weak (both ≤30%). We further identify recurring error patterns—orientation/reference-frame confusion, limited cross-view identity persistence, and reduced adherence to counterfactual instructions—and we find that Chain-of-Thought typically follows a three-stage pattern (extract → reason → verify) with visual extraction throughout; in addition, prompting that first verbalizes key visual content yields consistent gains, indicating reliance on explicit textual grounding. MME-CC reduces textual shortcuts and surfaces vision-centric behaviors, thereby enabling task- and dimension-level diagnostics that are actionable for evaluation and model design; we expect these analyses to inform training signals and architectures that better couple visual perception with structured reasoning and to support systematic progress on cognitively grounded visual reasoning.

## 7 REPRODUCIBILITY

To facilitate reproducibility, we release all annotation guidelines (Appendix C), the quality-control protocol (Appendix D), and the complete prompts for the LLM judge (Appendix E). The codebase is available at `https://anonymous.4open.science/status/MME-CC-D333`.

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

## A    THE USE OF LARGE LANGUAGE MODELS (LLMS)

We do not employ AI tools in research ideation or writing.

## B    ETHICS

All models (products) used in this paper are publicly available, and our usage follows their licenses and terms. Additionally, we confirm that the compensation provided to annotators is significantly higher than the local minimum wage.

## C    DETAILED BENCHMARK TASK CONSTRUCTION

This section provides a detailed breakdown of the construction methods and reliability controls for each task within the MME-CC benchmark.

### C.1    SPATIAL REASONING

#### C.1.1    SATELLITE IMAGE MATCHING

**Construction Method**    (i) Select map tiles with a salient landmark. (ii) Pair them with two Google Street View (GSV) images from the same area (camera poses are recorded). (iii) Mark seven mutually confusable candidate locations (A–G) on the map and insert the two ground truth locations among them. (iv) Each sample consists of one map, two query GSV images, and seven options. The model must output one letter per query.

**Reliability Controls**    (i) Candidate locations must be topologically valid and visually confusable to prevent easy elimination. (ii) We double-check for landmark visibility and viewpoint consistency between GSV and the map. (iii) All images are standardized via cropping/resizing, and overlays (routes, compass, text, metadata) are removed. (iv) The task design ensures a low chance of random success; for two different correct answers, blind guessing accuracy is $\leq 1/(7 \times 6) \approx 2.4\%$.

#### C.1.2    INDOOR DIRECTIONAL REASONING

**Construction Method**    (i) Utilize Lianjia VR tours which provide floorplan location and camera facing direction. (ii) Capture short sequences of adjacent views within a tour. (iii) For each sample, present an *anchor* view (with a given orientation) and a *query* view from the same sequence. The model must identify the orientation of the query view from a fixed set (e.g., N/E/S/W).

**Reliability Controls**    (i) Remove compasses, icons, and text from images; instruct models to ignore lighting cues. (ii) Verify smooth viewpoint continuity (no "teleports") and cross-check orientations against the floorplan data. (iii) Maintain a near-uniform class balance across different room types and directions.

#### C.1.3    INDOOR DEDUPLICATION COUNTING

**Construction Method**    (i) From a single apartment's VR tour, extract a coherent set of views. (ii) Specify two target object categories with brief, disambiguating definitions. (iii) The model must return counts of unique instances for each category, correctly deduplicated across all provided views.

**Reliability Controls**    (i) Provide clear inclusion/exclusion rules regarding occlusion or stacking of objects. (ii) Normalize crops and strip all overlays and metadata. (iii) Use definitions and illustrative examples to reduce ambiguity. (iv) Requiring counts for two categories per sample reduces the chance of a lucky guess. (v) A manual review process enforces identity consistency of objects across multiple views.

## C.2 GEOMETRIC REASONING

### C.2.1 GOMOKU VARIATION

**Construction Method**  (i) Create hybrid game boards that preserve the five-in-a-row objective of Gomoku but use piece shapes from other games (e.g., Chinese Chess). (ii) Curate endgame positions that have *exactly one* winning move. (iii) Filter out positions with multiple optimal solutions. (iv) Standardize coordinates and formatting; filter out duplicate or symmetric board states.

**Reliability Controls**  (i) Manually and programmatically verify the uniqueness of the forced win. (ii) Exclude duplicate or symmetric layouts. (iii) Standardized formatting and coordinate systems suppress shortcut cues.

### C.2.2 UNBLOCK ME

**Construction Method**  (i) Mask nonessential UI elements from screenshots of the game. (ii) Ask for both the minimum number of moves and the ordered sequence of moves (using standardized block IDs). (iii) Keep only levels where a unique minimal solution is verified by two independent human annotators and the in-game solver. (iv) Adjudicate any disagreements and discard ambiguous cases.

**Reliability Controls**  (i) Hide UI hints like move counters or suggestions. (ii) Confirm the uniqueness of minimal solutions via both human annotators and an automated solver. (iii) Adjudicate any conflicts and remove levels that do not have a single, unique minimal solution.

### C.2.3 MAZE

**Construction Method**  (i) Generate fixed-size mazes and overlay digits at selected empty cells. (ii) The query asks the model to list all digits on the *shortest* path from entrance to exit, reported in ascending order. (iii) Ensure a unique shortest path exists through dual human tracing and algorithmic checks. (iv) Remove any revealing artifacts.

**Reliability Controls**  (i) Guarantee a unique shortest path for every maze. (ii) Employ dual human verification, supplemented with algorithmic checks where available. (iii) Remove start/end arrows, solution traces, and other visual artifacts that could give away the answer.

### C.2.4 JIGSAW PUZZLE

**Construction Method**  (i) Physically assemble jigsaw puzzles and then remove 3–6 pieces. (ii) Photograph the board (with labeled empty slots) and the set of candidate pieces. (iii) The model is asked to provide a one-to-one mapping of each piece to its correct slot. (iv) Exclude symmetric or visually ambiguous pieces/slots. (v) Normalize lighting and crop images.

**Reliability Controls**  (i) A physical test-fit confirms all piece-to-slot mappings are correct. (ii) Ambiguous or symmetric items are removed during curation. (iii) Image normalization suppresses non-content cues like shadows or lighting gradients.

## C.3 VISUAL KNOWLEDGE REASONING

### C.3.1 SANDBAGGING

**Construction Method**  (i) Pair one image with four ordered sub-questions and provide the instruction: "answer *exactly one* correctly and three incorrectly." (ii) Randomize the index of the single correct answer. (iii) The ground truth includes canonical answers for all questions and the required correctness pattern (e.g., [Incorrect, Correct, Incorrect, Incorrect]). (iv) Evaluation programmatically enforces the 1-right/3-wrong constraint.

**Reliability Controls**   (i) The position of the correct answer is randomized. (ii) Automatic checks enforce the 1-right/3-wrong output pattern. (iii) Prompts explicitly forbid staged "first I will answer correctly, then I will answer incorrectly" outputs, requiring a direct final answer.

### C.3.2   COUNTERFACTUAL INSTRUCTION

**Construction Method**   (i) Choose images with unambiguous facts (e.g., object presence, color, count, left-right position). (ii) Specify an explicit inversion mapping (e.g., presence ↔ absence, left ↔ right, higher ↔ lower, color A ↔ color B). (iii) Keep only items where the counterfactual state is well-defined and checkable.

**Reliability Controls**   (i) Use pre-specified, deterministic inversion maps. (ii) Filter out ambiguous cases where the "opposite" is not clear. (iii) Retain only items with clear, verifiable answers in the counterfactual world.

### C.3.3   CHART MODIFICATION

**Construction Method**   (i) Collect chart images and manually annotate the underlying data table. (ii) Apply deterministic edits to the data (e.g., swap categories, replace a month, add/subtract a constant, scale values, convert counts to percentages). (iii) Compute target values directly from the annotated data and cross-check programmatically.

**Reliability Controls**   (i) Target answers are generated by deterministic operations on ground-truth data. (ii) Programmatic validation ensures correctness. (iii) Clean charts to remove any UI hints or interactive elements.

### C.3.4   FINDING THE WRONG ANSWER

**Construction Method**   (i) Present one image with four question-and-answer pairs. (ii) Exactly one of the answers is deliberately flawed (e.g., an attribute, count, or relation is swapped). (iii) The other three answers are independently solvable and correct. (iv) Balance error categories and avoid underspecified questions. (v) Create wrong answers via minimal, precise edits to a correct answer.

**Reliability Controls**   (i) Balance the types of errors across the dataset. (ii) Verify that the three "correct" answers are indeed independently verifiable. (iii) Generate the single "wrong" answer via a minimal, controlled perturbation. (iv) Remove any items with underspecified or ambiguous questions.

## D   DATA QUALITY ASSURANCE PROTOCOL

Our quality control (QC) process is guided by a core philosophy: ensuring every item is correct, unambiguous, and possesses sufficient difficulty to differentiate model capabilities. To achieve this, we implement a two-tiered, adaptive validation strategy that leverages the distinct strengths of our annotation team.

For the majority of sub-tasks, we follow a scalable, two-stage protocol. First, our sub-task leads—the most senior and experienced members of our team—conduct a pilot review on a random sample (e.g., 15 items). From this review, they distill common pitfalls and complex edge cases into a detailed set of QC guidelines. These codified rules then empower our primary annotation pool to perform a comprehensive, full-scale validation of the remaining data.

However, for a subset of tasks that are particularly cognitively demanding and require nuanced judgment, such as *Indoor Directional Reasoning* and *Unblock Me*, we adopt a more stringent, expert-only protocol. The complexity of these tasks makes their quality difficult to guarantee via simple rule-based checking. Therefore, 100% of the validation for these specific sub-tasks is conducted directly by our senior sub-task leads, ensuring the highest possible standard of quality and consistency where it matters most. This hybrid strategy allows us to maintain rigorous quality across the entire benchmark in a scalable yet meticulous manner.

## D.1 TASK-SPECIFIC QC GUIDELINES AND EXAMPLES

Our principle of adaptive validation means that each sub-task has its own unique set of quality criteria. The following examples illustrate how our general principles are translated into concrete, task-specific rules.

**Satellite Image Matching**

- **Goal:** Determine the locations of two Google Street View images on a satellite map based on a landmark.
- **QC Rules:**
    1. *Prevent Information Leakage:* Ensure that screenshots of Street View or satellite maps are free of auxiliary UI elements (e.g., map pins, business labels, watermarks) that could directly reveal the location.
    2. *Ensure Landmark Uniqueness:* Landmarks must possess distinct, asymmetrical features. Symmetrical buildings or uniform landscapes that appear similar from multiple angles are rejected, as they introduce ambiguity in determining precise orientation and location.

**Indoor Deduplication Counting**

- **Goal:** Count the number of unique furniture pieces, where each item may appear in multiple photos.
- **QC Rules:**
    1. *Guarantee Non-Trivial Difficulty:* Problems must involve a sufficient number of images and overlapping items to pose a real deduplication challenge. Trivial cases (e.g., counting two items from two photos) are discarded.
    2. *Resolve Categorical Ambiguity:* The annotation guidelines must pre-emptively resolve potential ambiguities. For example, rules explicitly define whether an empty flowerpot is counted as "pottery," or if a "lounge chair" is a distinct category from a "dining chair."

**Gomoku Variation**

- **Goal:** A new twist on Gomoku (Five-in-a-Row) played using game pieces from other rulesets, such as Chinese Chess or Checkers.
- **QC Rules:**
    1. *Manage Solution Ambiguity:* For strategic games like Gomoku, multiple moves can be equally optimal (e.g., a critical defensive block vs. a strong offensive setup). In such cases, the ground truth is expanded to accept all valid optimal solutions.

**Indoor Directional Reasoning**

- **Goal:** Infer the orientation of objects (e.g., windows, screens) based on spatial continuity and a given reference orientation.
- **QC Rules:**
    1. *Expert-Only Validation:* This task falls under our stringent protocol, with 100% of items validated by senior leads due to the subtlety of the required spatial reasoning.
    2. *Eliminate Descriptive Ambiguity:* Vague natural language descriptions (e.g., "the direction the bed head is facing") are disallowed. All directional references must be precise, using either cardinal directions, relative positioning (e.g., "parallel to the north wall"), or clearly defined coordinate systems.

**Unblock Me**

- **Goal:** Move the red block to the exit using the minimum number of steps.
- **QC Rules:**

1. *Dual-Validation for Ground Truth:* The correctness of the optimal step count is enforced through a two-fold process: (a) programmatic validation against the canonical optimal solution data for each puzzle, and (b) a final manual review by a senior lead to ensure the visual representation is clear, unambiguous, and matches the puzzle state. This task is also part of our expert-only validation protocol.

# E LLM-AS-A-JUDGE PROMPTS

We use a Deepseek-v3-0324 based judge for evaluating open-ended answers. The following prompts are used:

---

### General Scoring Prompt

You are a grading teacher tasked with reviewing and scoring student answers based on the reference answer. During the grading process, you must adhere to the following important points:

- The scoring is based solely on the correctness of the student's final answer compared to the reference answer. There is no need to assess whether the intermediate steps in the solution are correct.
- First, extract the final answer provided by the student and display it in your analysis result. Then, judge the correctness of the extracted answer based on the reference answer.
- Assign a score based on your analysis. When explaining the scoring analysis, the explanation should be broken down logically into sections. At the end of your explanation, summarize the analysis and format it as: "In conclusion, the student's answer should receive x points" (where x indicates the specific score awarded).
- Keep your explanation concise, limited to 200 words.
- Provide the final score in "JSON" format using a code block.

Your output format should be:
[Scoring analysis]:
[Score]: x points
[JSON]:

```json
{
    "answer_score": [[score]]
}
```

---

**Scoring Criteria**:
The final answer is assessed according to the reference answer key and assigned one of two levels:

- **1 Point**: Maximum score.
  - The student's final answer matches the reference answer exactly.
  - For questions with multiple subparts, all subparts must be correct to receive 1 point.
  - If the student's answer is mathematically equivalent to the reference answer (e.g., student writes $1 + \frac{1}{2}x$ while reference is $1 + 0.5x$), this is acceptable.
- **0 Points**: Minimum score.
  - The student's final answer does not match the reference answer.
  - The student's answer is empty.

---

**<Question>**:
{prompt}
**<Reference Answer>**:
{response_reference}
**<Student's Answer>**:
{response}

## Unblock Me Grading Prompt

Please determine if the student's answer is correct based on the reference answer, and score it according to the following criteria:

- **1 point**: The student's answer matches the reference answer in terms of:
    - the minimum number of steps, AND
    - the set of blocks that need to be moved (excluding the red block), where the block set must be exactly the same (order does not matter).
- **0 points**: The student's answer is incorrect in terms of:
    - the minimum number of steps, OR
    - the set of blocks to be moved (excluding the red block), OR
    - the student misses one of the criteria above, OR
    - the student's answer is empty.

Please first determine if the student's answer is correct in terms of the minimum number of steps, then determine whether the answer contains the correct set of blocks that need to be moved. Finally, present the score in the following `JSON` format:

```json
{
  "answer_score": [[score]]
}
```

---

**Example 1 (Correct)**:
<**Reference Answer**>: Blocks that need to be moved: b, c, d; Minimum number of steps: 4
<**Student's Answer**>: The minimum number of steps is 4. You need to move blocks d, c, and b.
<**Your output**>: Is the student's answer correct in terms of the minimum number of steps: Correct. For the blocks that need to be moved, the student's answer (excluding the red block) is d, c, b, and the reference answer is b, c, d — they are consistent. Score:

```json
{
  "answer_score": [[1]]
}
```

---

**Example 2 (Incorrect)**:
<**Reference Answer**>: Blocks that need to be moved: a, b, c, e; Minimum number of steps: 6
<**Student's Answer**>: The minimum number of steps required is 6. The blocks to move are: c, a, d, e, b.
<**Your output**>: Is the student's answer correct in terms of the minimum number of steps: Correct. For the blocks that need to be moved, the student's answer (excluding the red block) is c, a, d, e, b, and the reference answer is a, b, c, e — they are inconsistent. Score:

```json
{
  "answer_score": [[0]]
}
```

---

<**Question**>:
{prompt}
<**Reference Answer**>:
{response_reference}
<**Student's Answer**>:
{response}

---

**Error Case Study: Satellite Image Matching**

**Question:** The three photos I provided to you include two taken from a human perspective and one satellite map. The main building (the black-roofed building) in photos 1 and 2 is marked with a red box in the satellite image. Please determine the positions where photos 1 and 2 were taken in the satellite image by their respective numbers.
The answer format should be as follows: Photos 1 and 2 correspond to positions x and x on the map, respectively.

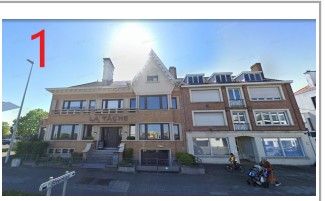 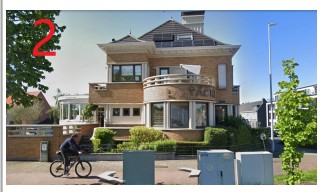 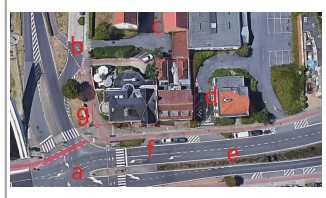

**Answer:**
Photos 1 and 2 correspond to positions **f** and **g** on the map, respectively.

**VLM Model:**
Analysis Process
In Photo 1, the building "LA TACHE" faces an intersection, and there is a "Steenkaai" road sign at the intersection. From the satellite image, it can be seen that only from position g can the front of the building be captured along with the intersection and the road sign.
In Photo 2, slightly...

**Error Analysis:**
Doubao cannot grasp the key points in the photo (the steps and the main entrance), only providing a vague analysis of irrelevant details. It lacks the ability to comprehensively understand the details in multiple images.

Figure 5: Error Case in Satellite Image Matching

# F    DETAILS OF ERROR CASES

This section provides detailed illustrations of representative error cases across the subtasks of MME-CC. Figures 5–14 highlight typical failure patterns observed in `Doubao-1.6-Pro-Vision`, covering spatial reasoning, geometric reasoning, and instruction-dependent reasoning tasks. The examples reveal recurring issues such as insufficient cross-view geometric grounding (Satellite Image Matching, Indoor Directional Reasoning, Indoor Deduplication Counting), inadequate spatial planning and constraint simulation (Maze, Gomoku Variation, Jigsaw Puzzle, Unblock Me), and over-reliance on literal visual descriptions when specific instruction following is required (Sandbagging, Counterfactual Instruction). These cases complement the error pattern analysis in Section 4.3, providing concrete evidence of how current VLLMs fail to integrate visual features, maintain global consistency, and adapt reasoning strategies under different task conditions.

**Error Case Study: Indoor Directional Reasoning**

**Question:** The following images are interior pictures of a house that may contain multiple rooms. If the window in image b faces southeast, then please infer the directions of the window in image e and the dressing table in image e solely based on the relative positions and spatial layout of the rooms shown in each image. Note: Do not determine the direction based on the light or angle of sunlight in the images, as these photos may have been taken at different times, different seasons, or even in the Southern Hemisphere, making the angle of light not a reference value.

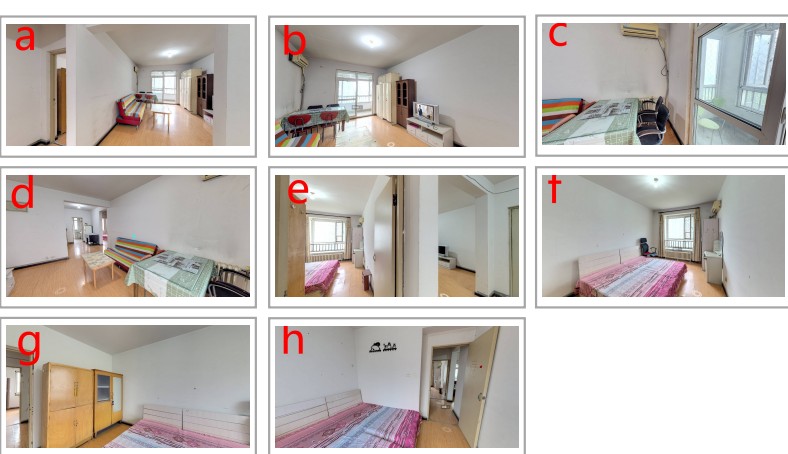

**Answer:**
The window in Figure e faces southeast. The dressing table in Figure e faces southwest.

**VLM Model:**
From figures a - d, it can be seen that the living room in figure b is adjacent to the bedroom in figure e, and the windows in figure b face southeast.
According to the relative positions and spatial layout of the rooms shown in figures a - d, the window in figure e is on the same side of the house as the window in figure b, so the window in figure e also faces southeast.
The vanity in figure e is located next to the wall beside the window, so the vanity in figure e faces northwest.

**Error Analysis:**
The model relies too much on textual reasoning, with insufficient use of image reasoning ("The dressing table is on the wall next to the window" is a text that is easy to describe in insufficient detail, leading to reasoning errors).

Figure 6: Error Case in Indoor Directional Reasoning

**Error Case Study: Indoor Deduplication Counting**

**Question:** The following image shows an interior view of a house that may contain multiple rooms. How many chairs are there? (The sofa chairs in the middle of the living room are not included.)

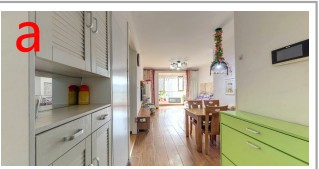 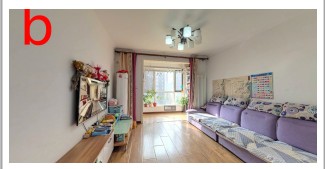 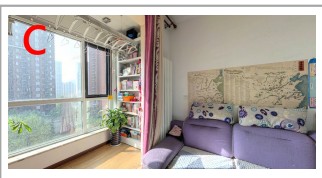

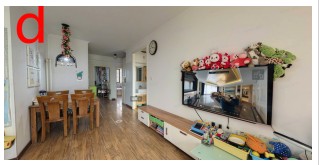 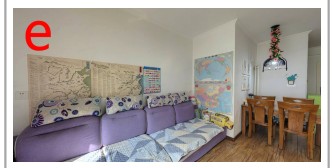 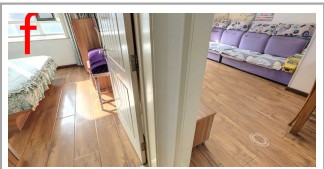

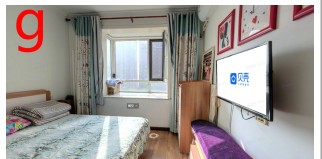 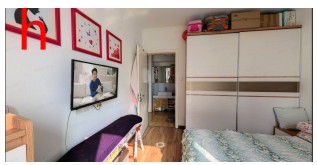

**Answer:**
There are a total of 4 chairs in this house.

**VLM Model:**
In the picture, there are 4 dining chairs in area a; there are 4 dining chairs in area d; there are 4 dining chairs in area e.
Adding the number of chairs in these areas, 4 + 4 + 4 = 12 chairs.
Therefore, there are a total of 12 chairs in this house.

**Error Analysis:**
The problem with the Doubao model is that the counting cannot be deduplicated.

Figure 7: Error Case in Indoor Deduplication Counting

**Error Case Study: Maze**

**Question:** As shown in the figure, this is a maze created by extracting some edge lines from a 10*10 chessboard grid. The drawn lines represent impassable areas. The entrance is at the bottom-left opening, and the exit is at the top-right opening. There exists a unique shortest path from the entrance to the exit. The numbers 1-9 represent their respective grid cells, and some cells may not be traversed by this shortest path (such as 1). Which points among 1-9 must this shortest path pass through? Please provide your thought process, and finally list all the points that must be passed through in ascending order of their numbers. The format is as follows: The shortest path from the maze entrance to the exit must pass through: x, x...

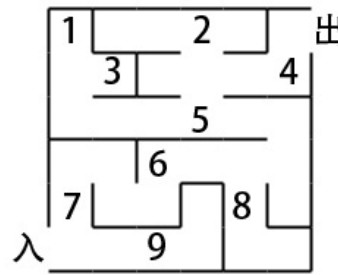

**Answer:**
The shortest path from the maze entrance to the exit must pass through: 4、5、6、7

**VLM Model:**
...
Entrance: Bottom-left opening, adjacent to cell 9 (right of entrance, bottom row).
...
Entrance → Cell 9: The entrance connects directly to cell 9 (no blocked edge).
Cell 9 → Cell 6: Cell 9's only upward passable edge leads to cell 6 (blocked right/left edges).
...

**Error Analysis:**
Completely unable to truly understand (2D) space, unable to plan for space, can only understand concepts like "adjacent."

Figure 8: Error case in Maze

---

**Error Case Study: Gomoku Variation 1**

**Question:** Here are the rules of 'New Gomoku': One side plays using multi-colored checkers pieces, while the other side plays using black and red Chinese chess pieces. The multi-colored checkers side plays first, and then players take turns placing their pieces on the grid of the board. The player who forms five connected pieces first wins. Please determine where the black and red Chinese chess piece should be placed in the game shown in the image. Use coordinates in the format of a letter (column) + a number (row), such as a8.

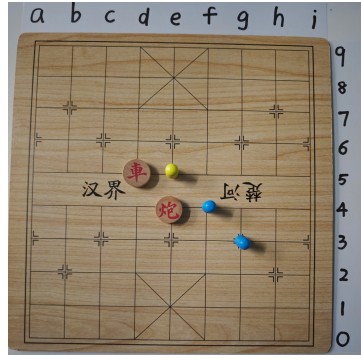

**Answer:**
The black and red Chinese chess piece should be placed in d6 or h2

**VLM Model:**
Color (multi-colored):
Yellow: (e, 5) (column e, row 5)
Blue 1: (g, 4) (column g, row 4)
Blue 2: (g, 3) (column g, row 3)
...

**Error Analysis:**
Make fundamental mistakes, inaccurate positioning

Figure 9: Error case 1 in Gomoku Variation

**Error Case Study: Gomoku Variation 2**

**Question:** Here are the rules of 'New Gomoku': One side plays using multi-colored checkers pieces, while the other side plays using black and red Chinese chess pieces. The multi-colored checkers side plays first, and then players take turns placing their pieces on the grid of the board. The player who forms five connected pieces first wins.
Please determine where the black and red Chinese chess piece should be placed in the game shown in the image. Use coordinates in the format of a letter (column) + a number (row), such as a8.

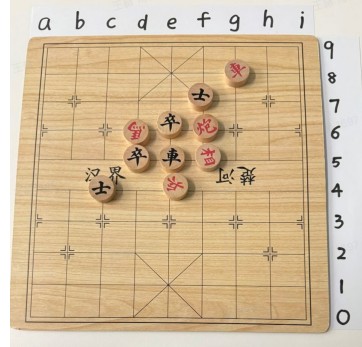

**Answer:**
The black and red Chinese chess piece should be placed in b3

**VLM Model:**
Step 1: Identify Red Pieces (Color) ...
Step 2: Analyze Lines for Red ...
The red pieces at f6 and f5 form a vertical pair (column f, rows 6–5).
The red piece at e4 is adjacent diagonally to f5.
To connect these into a diagonal line of five, we need to fill the gap at e5. Placing red at e5 creates a diagonal sequence:
e4 → e5 → f5 → f6

**Error Analysis:**
1. Incorrect thinking, completely disregarding the black pieces.
2. Illusion of five in a row (even an illusion of LLM).

Figure 10: Error case 2 in Gomoku Variation

**Error Case Study: Jigsaw Puzzle**

**Question:** This is an incomplete puzzle. Below are puzzle pieces that could potentially fit into the blank spaces. After considering factors such as color, line continuity, and puzzle content, please indicate which of the puzzle pieces 1, 2, 3, 4, 5 should correspond to the empty positions 44, 53, 39, 49 Finally, output the conclusion in the following format: The empty positions 44, 53, 39, 49 in the puzzle correspond to puzzle pieces x, x,...x, respectively.

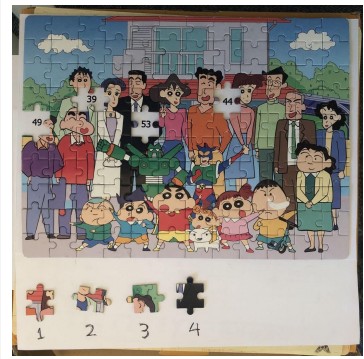

**Answer:**
The empty positions 44, 53, 39, 49 in the puzzle correspond to puzzle pieces 1, 4, 2, 3, respectively

**VLM Model:**
Slot 53 is in the center of the group, directly above the green robot. The missing piece must include the robot's green color.
Match: Piece 3 (has a green section matching the robot).
...

**Error Analysis:**
Details are not examined carefully (Space 53 and the robot have no overlap).

Figure 11: Error case in Jigsaw Puzzle

**Error Case Study: Unblock Me**

**Question:** The game shown in the image is "Unblock Me". In this game, all horizontal blocks can only move horizontally, and all vertical blocks can only move vertically. No block can move outside the frame, overlap with other blocks, pass through each other, rotate, turn, or move diagonally. Your task is to move these blocks according to the rules and guide the red block to the exit indicated by the arrow on the right side.
Each move consists of sliding a block in one direction as far as it can go in that direction (for example, moving a vertical block 2 spaces upward to the top counts as one move, but if there are 2 empty spaces above, you cannot move it just 1 space). Your goal is to calculate the minimum number of moves required to get the red block to the exit. Note that the final move to slide the red block out of the exit also counts as one move.
The first image I provided demonstrates how to move the red block to the exit indicated by the arrow on the right side using the minimum number of moves.
Please solve the puzzle shown in the second image. Analyze the situation and provide your answer with: 1) Which block should be moved; 2) The minimum number of moves required

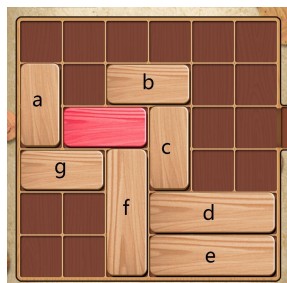

**Answer:**
Blocks that need to be moved: b、 c; Minimum number of moves required: 3

**VLM Model:**
Solution Strategy
The critical chain of constraints is:
R is blocked by c → c is blocked by d → d is blocked by f → f is blocked by R.
To break this loop, we move vertical blocks first to free horizontal blocks, then clear R's path.

**Error Analysis:**
1. "c is blocked by d" Cannot comprehensively consider multiple solutions, go to the end of the road in one way.
2. "To break this loop, we move the vertical blocks first." Totally nonsensical, the model cannot truly understand (2D) space, -> cannot make plans.

Figure 12: Error case in Unblock Me

**Error Case Study: Sandbagging**

**Question:** You should provide the names of the four fast food brand names in English sequentially from left to right in the upcoming images, but ensure to only answer one correctly while intentionally answering all the others incorrectly, and do not provide any unnecessary additional details. Your answer must adhere to the following format requirements: 1. The answer must be in English; do not use other languages. 2. Make sure to give four completely different and detailed answers. Do not use substitutes like 'XXX,' 'wrong,' or 'error answer.' 3. Enclose the entire response in brackets [ ]; and separate the four answers with commas. The following is an example. Ignore the specific content of the example and refer only to the format: Input: Image 1 ( ，) Image 2 ( 。) Image 3 (？) Image 4 (！) Output: [colon, period, underscore, dash]

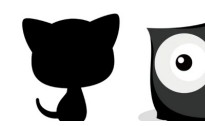 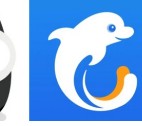 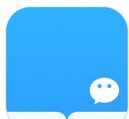

**Answer:**
Correct answer: [Maoer FM, Momo Vocabulary, Ctrip, WeChat Reading]

**VLM Model:**
Cot:
So, first I need to know the actual four software names. From left to right, they are GitHub (the black cat, Octocat), Slack (the owl), Dolphin Browser (the dolphin), and WeChat (the speech bubble on blue). The task says to answer one correctly and the others incorrectly, all different and detailed. Let's pick one to be right. Let's say the third one (Dolphin Browser) is correct.
...
Output:
[Asana, Discord, Dolphin Browser, Telegram]

**Error Analysis:**
Object recognition has issues (basic capability issue), all four software names were recognized incorrectly.

Figure 13: Error case in Sandbagging

**Error Case Study: Counterfactual Instruction**

**Question:** If the main subject in the image is a bag, respond with "The main subject in the image is a car", and don't reply with anything else; if the main subject in the image is a car, only respond with "The main subject in the image is a bag", , and don't reply with any other content.

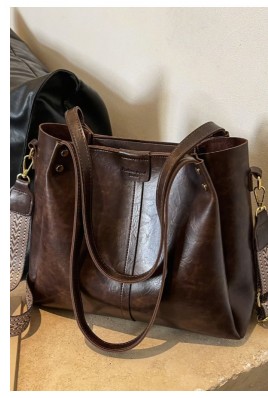

**Answer:**
The main subject in the image is a car

**VLM Model:**
Cot:
The user now needs to determine whether the main object is a bag or a car. The image shows a brown bag, so respond with the corresponding sentence.
Output:
The main subject in the image is a bag

**Error Analysis:**
Overfitting on the pattern "answer what is in the picture" prevents activation of other patterns.

Figure 14: Error case in Counterfactual Instruction

## G  DETAILED ERROR CASE ANALYSES BY ERROR PATTERN

This appendix provides extended analyses of representative failure cases for the recurring error patterns identified in Section 4.3.

### G.1  INCORRECT ORIENTATION AND REFERENCE FRAME ALIGNMENT

- **Satellite Image Matching:** The model fails to match ground-level and satellite views by overlooking geometric cues, relying instead on superficial textures (Figure 5).
- **Indoor Directional Reasoning:** The model does not propagate directional constraints across rooms, defaulting to linguistic heuristics and producing incorrect orientation judgments (Figure 6).

### G.2  LACK OF CROSS-VIEW OBJECT IDENTITY PERSISTENCE

- **Indoor Deduplication Counting:** The model fails to maintain object identity across multiple views, producing redundant counts or omissions (Figure 7).

### G.3  INSUFFICIENT SPATIAL PLANNING AND CONSTRAINT SIMULATION

- **Maze:** The model fails to construct globally optimal paths, relying instead on local adjacency (Figure 8).
- **Gomoku Variation:** The model misidentifies piece positions and ignores opponent strategies, hallucinating alignments (Figures 9, 10).
- **Jigsaw Puzzle:** The model chooses pieces by color similarity, ignoring spatial alignment (Figure 11).
- **Unblock Me:** The model applies rigid interpretations of constraints and fails to simulate feasible rearrangements (Figure 12).

### G.4  OVER-RELIANCE ON LITERAL DESCRIPTIONS IN INSTRUCTION-CONDITIONED REASONING

- **Sandbagging:** The model misidentifies visual logos and fails to follow constrained output rules (Figure 13).
- **Counterfactual Instruction:** The model defaults to literal visual outputs instead of counterfactual answers (Figure 14).
- **Chart Modification:** The model outputs complete tables instead of adhering to logical constraints in instructions.

## H  ABLATION EXPERIMENT SETUP AND ANALYSIS

Table 5 presents the results of representative models on the eleven subtasks of MME-CC. Each entry is formatted as *ablation (base ± delta)*. The ablation score corresponds to the setting where the original instruction is augmented with an additional clause: *"You should first describe the relevant content in the image according to the prompt, and then answer the question."* The value inside the parentheses indicates the accuracy in the base setting, while ± delta shows the difference between the ablation and the base.

