# OpenReview forum: "MME-CC: A Challenging Multi-Modal Evaluation Benchmark of Cognitive Capacity"
_ICLR.cc/2026/Conference — ICLR 2026 Conference Withdrawn Submission_

### Official Review · Reviewer_kJMT · 2025-10-22

**Soundness:** 3
**Presentation:** 2
**Contribution:** 2
**Rating:** 4
**Confidence:** 5

**Summary:**

This paper introduces MME-CC, a comprehensive vision-grounded benchmark designed to evaluate the cognitive capacity of MLLMs across three reasoning dimensions: spatial, geometric, and visual knowledge reasoning.
It includes 11 representative tasks and 1173 human-verified samples, ensuring high quality through multi-stage annotation, review, and model-based filtering.
Experiments on 16 MLLMs reveal that closed-source models like Gemini-2.5-Pro (42.66%) outperform open-source ones, but all struggle with spatial and geometric reasoning (below 30%).
The study identifies common error patterns, including orientation errors, identity inconsistency, and poor counterfactual reasoning, and finds that MLLMs typically follow a three-stage reasoning chain (extract → reason → verify), highlighting the need for cognitively grounded multimodal evaluation.

**Strengths:**

+ Evaluated 16 MLLMs, covering major frontier models.
+ All collected test cases undergo human verification (at least 2 annotators).

**Weaknesses:**

- The benchmark contains 11 tasks, categorized into Spatial Information, Geometric Information, and Visual Knowledge Reasoning. The major weakness in this paper is that the authors do not address the “why” issue. Why do you pick these three major directions? Are they essential in human cognition (are they prerequisites of other downstream tasks)? Or are they key performance indicator of some human abilities? Why do you pick the 11 tasks for the three directions? Are they sufficiently representative? In conclusion, the authors should answer why other researchers should use your benchmark instead of others, and answer how the community can position your benchmark. This point is highly related the last point in my weaknesses section: I do not think the current paper has done enough literature review.
- Since the paper’s main contribution is the proposed benchmark, the construction process is very important. More information about the annotation process should be included. Where did you recruit the annotators? What is the requirement for qualification (e.g., years of experience in AI research field)? How long did each of them take for the annotation task? Do you provide reasonable compensation (how much dollars per hour)? Each test case is checked by two annotators (from Line 210-213), what is their agreement rate?
- Another weakness is the single model used for LLM-as-a-Judge. Although the authors provide a small-scale experiment on human agreement (95%), a more reliable way is to use multiple LLMs-as-Judges. Single model may have biases on language style, sentence length, etc.
- The current paper is missing plenty of related papers to discuss (a list alphabetically ordered by first author’s last name is provided below). [9, 14] use Raven’s Progressive Matrices and [4] uses Factor-Referenced Cognitive Test. [10] uses the Logic Test from the Chinese Civil Service Examination. [9, 12] use mental rotation tests. [1, 6] are very related to spatial reasoning. These benchmarks focus more on the vision part, not like MathVista focusing mainly on the text domain as you mention in the introduction. Many of them are also grounded to cognitive science. I truly think that you need to discuss these papers. Moreover, [2, 4] can automatically generate new test cases, not limited to static benchmarks.
1. Cheng, A.-C., Yin, H., Fu, Y., Guo, Q., Yang, R., Kautz, J., Wang, X., and Liu, S. Spatialrgpt: Grounded spatial reasoning in vision-language models. Advances in Neural Information Processing Systems, 37, 2024.
2. Feng, Y., Xu, Z., Jiang, F., Li, Y., Ramasubramanian, B., Niu, L., Lin, B. Y., and Poovendran, R. Visualsphinx: Large-scale synthetic vision logic puzzles for rl. arXiv preprint arXiv:2505.23977, 2025.
3. Fu, X., Hu, Y., Li, B., Feng, Y., Wang, H., Lin, X., Roth, D., Smith, N. A., Ma, W.-C., and Krishna, R. Blink: Multimodal large language models can see but not perceive. In European Conference on Computer Vision, pp. 148–166. Springer, 2024.
4. Huang, J. T., Dai, D., Huang, J. Y., Yuan, Y., Liu, X., Wang, W., Jiao, W., He, P., Tu, Z., and Duan, H. Human cognitive benchmarks reveal foundational visual gaps in mllms. arXiv preprint arXiv:2502.16435, 2025.
5. Kamath, A., Hessel, J., and Chang, K.-W. What’s “up” with vision-language models? investigating their struggle with spatial reasoning. In Proceedings of the 2023 Conference on Empirical Methods in Natural Language Processing, pp. 9161–9175, 2023.
6. Li, C., Zhang, C., Zhou, H., Collier, N., Korhonen, A., and Vulic, I. Topviewrs: Vision-language models as top-view ´ spatial reasoners. In Proceedings of the 2024 Conference on Empirical Methods in Natural Language Processing, pp. 1786–1807, 2024.
7. Li, Y., Gao, Q., Zhao, T., Wang, B., Sun, H., Lyu, H., Hawkins, R. D., Vasconcelos, N., Golan, T., Luo, D., and Deng, H. Core knowledge deficits in multi-modal language models. In Forty-second International Conference on Machine Learning, 2025b.
8. Liu, F., Emerson, G., and Collier, N. Visual spatial reasoning. Transactions of the Association for Computational Linguistics, 11:635–651, 2023.
9. Song, W., Li, Y., Xu, J., Wu, G., Ming, L., Yi, K., Luo, W., Li, H., Du, Y., Guo, F., et al. M3gia: A cognition inspired multilingual and multimodal general intelligence ability benchmark. arXiv preprint arXiv:2406.05343, 2024.
10. Song, Y., Ou, T., Kong, Y., Li, Z., Neubig, G., and Yue, X. Visualpuzzles: Decoupling multimodal reasoning evaluation from domain knowledge. arXiv preprint arXiv:2504.10342, 2025.
11. Rahmanzadehgervi, P., Bolton, L., Taesiri, M. R., and Nguyen, A. T. Vision language models are blind: Failing to translate detailed visual features into words. arXiv preprint arXiv:2407.06581, 2024.
12. Ramakrishnan, S. K., Wijmans, E., Kraehenbuehl, P., and Koltun, V. Does spatial cognition emerge in frontier models? In The Thirteenth International Conference on Learning Representations, 2025.
13. Rudman, W., Golovanevsky, M., Bar, A., Palit, V., LeCun, Y., Eickhoff, C., and Singh, R. Forgotten polygons: Multimodal large language models are shape-blind. arXiv preprint arXiv:2502.15969, 2025.
14. Zhang, Y., Bai, H., Zhang, R., Gu, J., Zhai, S., Susskind, J., and Jaitly, N. How far are we from intelligent visual deductive reasoning? In The First Conference on Language Modeling, 2024.
15. Zhao, H. H., Zhou, P., Gao, D., and Shou, M. Z. Lova3: Learning to visual question answering, asking and assessment. Advances in Neural Information Processing Systems, 37, 2024.

**Questions:**

1. Gemini-2.5-Pro filtering: it seems that this model completes many tasks (Line 199-201). More details (like the prompt) should be given. There are some questions: since Gemini-2.5-Pro it is very strong, cases it can easily solve can be quite hard for smaller and weaker models. Maybe it is not a good idea to directly remove these easy ones. In the previous stage, you mention that Doubao and Gemini are used together. Why does this stage only include Gemini?
2. Table 3 caption: “all task information is derived from images rather than text.” How can we appropriately understand this claim? I think part of your task information is also from text, especially in the visual knowledge reasoning.
3. LLM-as-a-judge and human agreement: you select 33 from each category. But there are 5 tasks in the geometric reasoning while 3 tasks in the other two. Will it be better if you select proportionally more for geometric reasoning (i.e., 55)? Which model’s responses are you evaluating, or the 99 responses are randomly selected among all 16 models?
4. Since Gemini is highly incorporated in the task design phase, will there be systematic biases that make Gemini the best-performing model in your benchmark?
5. Since there is task size imbalance, do you do weighted average of the overall performance? Otherwise, the performance may be biased to the test containing the most cases.
6. Line 305-308: “This trend indicates that longer Chain-of-Thought reasoning chains provide additional opportunities for iterative verification of recognition outcomes and intermediate inferences, thereby contributing to improved problem-solving in complex scenarios.” Do you do correlation analysis on CoT length and the performance? Could you provide quantitative analysis here?
7. Is the “Extract → Reason → Verify” pattern summarized from Doubao can also be observed in other models?

Minor suggestions and typos:
1. Fig. 1 is too small to read.
2. Seems that you used \vspace commands in page 5 around the section titles of 3, 3.1, and 3.2. This violates ICLR style requirements. Please modify.

---

### Official Review · Reviewer_NYQr · 2025-10-27

**Soundness:** 2
**Presentation:** 3
**Contribution:** 2
**Rating:** 4
**Confidence:** 4

**Summary:**

This paper introduces MME-CC, a benchmark aimed at evaluating vision-grounded cognitive capacity of MLLMs. It organizes 11 visual reasoning tasks into three categories and provides fine-grained analyses across 16 representative models. The benchmark emphasizes visual reasoning independent of textual bias, highlighting current model weaknesses in spatial and geometric reasoning.

**Strengths:**

1. The paper is clearly organized and easy to follow. The benchmark taxonomy and pipeline descriptions are detailed and transparent.
2. MME-CC includes a diverse set of vision-centric reasoning tasks, systematically categorized by cognitive dimensions (spatial, geometric, visual-knowledge).
3. The paper evaluates 16 SOTA MLLMs, with thoughtful analyses of reasoning performance, scaling trends and error types.

**Weaknesses:**

1. While MME-CC is well-engineered, it largely repackages existing vision reasoning types under a new taxonomy. Prior works also emphasize visual reasoning or multimodal cognition. Compared with benchmarks like ZeroBench or MMStar, the main advance seems to be categorization and dataset curation rather than a fundamentally new evaluation concept.
2. The paper claims MME-CC focuses on “vision-based cognitive capacity” and is “language-independent,” but many tasks still depend on textual prompts for context and guidance. There is no ablation study of different prompt formulations or linguistic dependencies. This weakens the claim that the benchmark isolates visual cognition from textual reasoning.
3. The relationship between the instruction "first describe image-related content and then answer the question" in improving model performance and the inherent visual reasoning ability is not completely separated.
4. What is the difference between the perceptual reasoning and cognitive capacity claimed in this article?

**Questions:**

See weaknesses

---

### Official Review · Reviewer_4UjK · 2025-10-29

**Soundness:** 2
**Presentation:** 2
**Contribution:** 1
**Rating:** 2
**Confidence:** 4

**Summary:**

They build a benchmark for probing the cognitive capacity of multimodal LLMs. It organizes 11 tasks into three dimensions—Spatial reasoning, Geometric reasoning, and Visual-knowledge reasoning—and evaluates 16 models on 1,173 human-curated items using an LLM-as-a-judge protocol

**Strengths:**

1) Eleven subtasks (e.g., Satellite-Map matching, Indoor dedup-counting, Maze, Unblock Me, Counterfactual) cover some breadth
2) Human-in-the-loop construction, expert-only validation for tricky subtasks, standardized post-processing, and model-based filtering to remove trivial/ambiguous items.

**Weaknesses:**

1. While the use of an LLM-as-a-judge (DeepSeek-V3-0324) offers scalability and speed in evaluation, relying on a single model for correctness judgment introduces the risk of systematic bias. The authors mention a 95% agreement with human annotators across 99 samples, but this sample size is relatively small given the dataset’s size (1,173 items). There is no report of inter-rater reliability metrics (e.g., Cohen’s Kappa) nor a cross-model evaluation with alternative judges (e.g., GPT-4, Claude, Gemini). Since the judgment task itself is non-trivial—particularly for open-ended or visually grounded responses—this dependence on one model weakens the robustness and fairness of the evaluation. A stronger setup would involve multi-judge voting, deeper human-model agreement audits, and consistency checks across different judge models.
2. The evaluation scheme assigns either 0 or 1 point per answer, regardless of whether the model partially solved a complex reasoning task. This approach fails to capture graded understanding—for example, a model that correctly deduces 3 out of 4 digits in a maze path receives the same score (0) as one that guesses entirely wrong. For compositional tasks like Unblock Me, Jigsaw Puzzle, or Deduplication Counting, this binary feedback ignores intermediate reasoning success and fails to reflect progress. Incorporating partial credit, component-wise scoring, or confidence-weighted metrics could make the benchmark more informative and better suited to tracking developmental trajectories in model capability.
3. The paper introduces an ablation where prompts are augmented with a clause: “First describe the relevant content in the image, then answer the question.” This leads to consistent performance gains across tasks—but the interpretation is unclear. It's possible that the performance boost stems from better alignment with the judge’s expectations, rather than genuine improvements in visual reasoning. For example, models may perform better simply because they mimic the reference answer format more closely. Without control experiments using human raters or multiple judge models blind to the ablation condition, it's hard to disentangle reasoning quality from format conformity. This undermines the strength of the ablation-based conclusion.
4. While MME-CC covers three well-chosen dimensions—spatial, geometric, and visual-knowledge reasoning—it omits several important cognitive domains relevant to human-like multimodal reasoning. Notably, it lacks tasks involving temporal reasoning (e.g., across frames in videos), physical simulation or causality (e.g., intuitive physics), and perspective-taking or theory of mind. These omissions mean the benchmark, while strong in breadth within its domains, still underrepresents the full spectrum of cognitive faculties that future AGI systems must master.

**Questions:**

Many vision-based cognitive tasks in real-world scenarios (e.g., physical reasoning, social understanding, embodied navigation) unfold over temporal sequences. How do the authors envision extending MME-CC to handle dynamic inputs, such as video or egocentric image streams? Would the same taxonomy (spatial / geometric / visual-knowledge) still apply, or would a new set of cognitive dimensions be needed?

---

### Official Review · Reviewer_twKy · 2025-11-01

**Soundness:** 3
**Presentation:** 2
**Contribution:** 2
**Rating:** 4
**Confidence:** 4

**Summary:**

This work presents a benchmark aimed at evaluating the cognitive abilities of frontier models in a multi-modal, visual understanding context. The benchmark comprises 11 reasoning tasks categorized within 3 taxonomical buckets. The authors conduct experiments over a suite of open- and closed- source models demonstrating the shortcomings of models when it comes to human-like cognitive and spatial reasoning, and the observable patterns of model behaviour.

**Strengths:**

1. This benchmark is a valuable contribution to the burgeoning domain of multimodal benchmarks evaluating spatial reasoning.
2. The problem annotation and review process is sound and clear.
3. The analysis of error patterns is convincing and a useful avenue of exploration.

**Weaknesses:**

1. The work makes no mention of other benchmarks in the domain, or compare against similar benchmarks, or benchmarks used for a subset of the tasks compiled in this work (for example, EmbSpatial, Space3D-Bench for spatial reasoning, PolyMATH for geometric reasoning, and the many VQA datasets). This work could benefit from a better demonstration of how the problems in this benchmark are comparable or superior to existing benchmarks collated, perhaps as a more comprehensive extension to Table 3.
2.  This work could benefit from a meta-analysis on the problem space, answering questions like
- 2.1 What makes improving cognitive reasoning central to evaluation and model design and how this affects general model abilities on other tasks
- 2.2 What makes this dataset particularly challenging in this space
3. The analysis of error patterns is anecdotal, and could benefit from metrics to back up the assertions as well as to demonstrate comparative distributions of error types. In general, the research questions could benefit from greater exploration to underscore the challenges posed by this dataset.

**Questions:**

Questions
1. According to line 750, the cognitively-demanding tasks were reviewed by a small set of experts. What was the rubric for validation, and what was the validation judgement process? How did the process account for bias, or disagreement when there is more than one judge assessing quality of the question?

Suggestions
1. This work would benefit from a review of other benchmarks in the spatial reasoning evaluation domain, as this is a recently-developed domain with various other works that are not referenced in this paper

---

### Note · Authors · 2025-11-25

I have read and agree with the venue's withdrawal policy on behalf of myself and my co-authors.